# A Novel Affordable and Reliable Framework for Accurate Detection and Comprehensive Analysis of Somatic Mutations in Cancer

**DOI:** 10.3390/ijms25158044

**Published:** 2024-07-24

**Authors:** Rossano Atzeni, Matteo Massidda, Enrico Pieroni, Vincenzo Rallo, Massimo Pisu, Andrea Angius

**Affiliations:** 1Center for Advanced Studies, Research and Development in Sardinia (CRS4), 09050 Pula, Italy; ratzeni@crs4.it (R.A.); ep@crs4.it (E.P.); massimo@crs4.it (M.P.); 2Department of Medical, Surgical and Experimental Sciences, University of Sassari, 07100 Sassari, Italy; mmassidda@uniss.it; 3Istituto di Ricerca Genetica e Biomedica (IRGB), Consiglio Nazionale delle Ricerche (CNR), Cittadella Universitaria di Cagliari, 09042 Monserrato, Italy; vincenzo.rallo@irgb.cnr.it

**Keywords:** cancer, somatic mutations, mutational patterns, mutational signatures, somatic variant detection, machine learning, precision medicine

## Abstract

Accurate detection and analysis of somatic variants in cancer involve multiple third-party tools with complex dependencies and configurations, leading to laborious, error-prone, and time-consuming data conversions. This approach lacks accuracy, reproducibility, and portability, limiting clinical application. *Musta* was developed to address these issues as an end-to-end pipeline for detecting, classifying, and interpreting cancer mutations. *Musta* is based on a Python command-line tool designed to manage tumor-normal samples for precise somatic mutation analysis. The core is a Snakemake-based workflow that covers all key cancer genomics steps, including variant calling, mutational signature deconvolution, variant annotation, driver gene detection, pathway analysis, and tumor heterogeneity estimation. *Musta* is easy to install on any system via Docker, with a Makefile handling installation, configuration, and execution, allowing for full or partial pipeline runs. *Musta* has been validated at the CRS4-NGS Core facility and tested on large datasets from The Cancer Genome Atlas and the Beijing Institute of Genomics. *Musta* has proven robust and flexible for somatic variant analysis in cancer. It is user-friendly, requiring no specialized programming skills, and enables data processing with a single command line. Its reproducibility ensures consistent results across users following the same protocol.

## 1. Introduction

The large amount of data coming from cancer genomes, via high-throughput next-generation sequencing (NGS) technology, has recently revolutionized the studies in oncology, enabling comprehensive analyses of somatic variants and addressing the need for affordable data analysis tools, oriented to their identification and interpretation [1]. In fact, both single nucleotide variants (SNVs) and small insertions and deletions (INDELs) play a critical role in the initiation and progression of cancer [2]. Unfortunately, the specific somatic mutations that drive tumor growth exhibit high variability across different cancer types, and even within individual tumors [3]. Hence, it is vital to accurately identify and interpret these mutations for comprehensive understanding of cancer genome characterization, clinical genotyping, and treatment strategies [4].

However, the accurate identification and interpretation of somatic mutations in cancer samples remain a major challenge due to the inherent complexity and heterogeneity of cancer genomes [5]. This complexity varies among different tumor types, and even among patients with the same cancer type [6]. Additionally, detecting low allele frequency variants present in a small fraction of cancer cells is difficult using conventional sequencing methods [5]. Sequencing artifacts, such as errors during library preparation, sequencing, or data processing, can lead to false-positive or false-negative results, affecting the accuracy of somatic mutation detection [7]. Furthermore, cross-contamination between tumor and normal tissues during sample collection, processing, or sequencing can introduce spurious results. Another challenge is intratumor heterogeneity, which refers to the extensive genetic diversity among different tumor cells within the same tumor. Intratumor heterogeneity can arise due to clonal evolution or spatial heterogeneity, and can impact the accuracy of somatic mutation calling and structural and functional interpretation of mutations [5,6].

To address these challenges, researchers have developed various bioinformatics tools for processing and analyzing cancer genomic data, each with its strengths and limitations [8]. These tools typically encompass multiple analysis steps, including read alignment, variant calling, variant annotation, detection of driver genes, pathway analysis, assessment of tumor heterogeneity, and deconvolution of mutational signatures [8].

However, using third-party software for detecting and analyzing mutations in cancer poses further challenges and problems, such as inaccuracies, difficulties in reproducibility, and limited clinical applicability [8]. The complex dependency trees and configuration requirements of these tools can often exacerbate these issues. All these aspects highlight the necessity of user-friendly and streamlined pipelines that offer reproducible, reliable and accurate results. Developing such pipelines is crucial for standardizing the analysis of cancer genomic data and ensuring result reproducibility across different datasets and research groups [8].

To overcome these limitations, we present Mutation and Somatic Tumor Analysis (*Musta*): a novel, affordable, and reliable end-to-end pipeline for detecting, classifying, and interpreting somatic mutations in cancer. *Musta* is designed to provide a standardized framework that offers accurate detection and comprehensive analysis of somatic variants while being at the same time user-friendly and cost-effective.

## 2. Results

*Musta* is currently used for cancer sample data analysis at the CRS4-NGS Core. Its reliability has been extensively tested on published data from “The Cancer Genome Atlas” repository. Each cohort counts hundreds of samples in distinct patients. Furthermore, *Musta* was also tested on published data from the genome archive of Beijing Institute of Genomics, with 23 samples of liver cancer from the same patient [9].The performance evaluations of *Musta* were conducted on a Dell server with dual Intel Xeon Gold 6238R processors, each featuring 28 cores at 2.2 GHz, a 38.5 M cache, and 128 GB of RAM. The server, configured with a total storage capacity of 40 TB in a RAID 5 setup, provided ample resources for robust analyses. The server runs CentOS Linux release 7.9.2009, with Docker version 20.10.11 and GNU Make version 4.3. During evaluations, each job in the Snakemake workflow efficiently utilized 4 out of the 56 available cores on the server.

### 2.1. Evaluating *Musta* in a Hepatocellular Carcinoma (HCC) Dataset

In the initial evaluation of *Musta*, we focused on its performance in analyzing the HCC dataset. A set of 23 tumor biopsies, along with a tumor-adjacent matched normal sample (N1), were sequenced at an average depth of 74.4× [9]. The samples were systematically distributed across the tumor tissue slice, categorized into four quadrants (A–D), with a central sample (Z1). Furthermore, the 23 sequenced samples were evenly distributed within the tumor, with 12 samples positioned on the periphery and 11 samples on the inside (see Figure 1a in [9]).

#### 2.1.1. Detection

To evaluate the performance of *Musta*’s Detection module on HCC datasets, we executed the *Musta* detect command with all variant callers enabled. The results were then compared with the findings of the original study [9]. To provide a more comprehensive representation and facilitate interpretation, data from comparative results are summarized in plots and diagrams in Figure 1 and Figure 2. Initially, a purely quantitative assessment is conducted by comparing the total numbers and counts of “pass” variants, along with the execution times for each variant caller and sample (Appendix A). Additionally, Appendix A includes data on the amount of non-swapped physical and virtual memory used, the number of MB read and written, and CPU usage time.

In the overall variant caller analysis, distinctive characteristics emerge, as clearly depicted by the plots for runtime (Figure 1a,b) and the number of somatic variants (Figure 1c,d). LoFreq stands out for its consistency and reliable results, featuring quick runtimes but potentially limited sensitivity due to a lower count of reported variants. MuSe, despite variable runtimes, contributes to a comprehensive overview with its moderate to high number of reported variants. Mutect2 proves stable in runtimes, achieving a balance between precision and sensitivity with a moderate count of reported variants. Strelka2 excels in quick execution, offering a high quantity of variants but requiring careful filtering. VarDict, albeit slower, provides an extensive set of variants, emphasizing the trade-off between time and results. VarScan2 demonstrates versatility with moderate runtimes and a balance between precision and sensitivity, offering a well-rounded choice. The consensus strategy prioritizes variants identified by multiple callers, ensuring a more conservative selection. However, when integrating permissive and restrictive callers, there is a potential drawback: the rejection of a significant portion of variants called by the permissive ones. It is important to note the variation in execution times: Strelka2, LoFreq, and VarScan2 complete their analysis in a few tens of minutes, Mutect2 and MuSe take about an hour and a half, and VarDict requires over 24 h. Excluding VarDict, the Detection module typically takes about 4 h to analyze a single sample.

In general, individual variant callers share more mutations with *Musta* than with other variant callers, confirming *Musta*’s superiority as an ensemble approach (see Figure 2a and Appendix A). Mutect2 and Strelka2 are exceptions, sharing a significant portion of the identified somatic mutations. Additionally, VarDict demonstrates the highest concordance with Strelka2. Let us now examine the number of somatic variants per sample, identified and validated by [9], and the comparison with *Musta*’s results (Appendix A). For a comprehensive validation, and to demonstrate the efficacy of the ensemble approach, we also compare the results of individual variant callers and of combinations of them. From the calculation of the root mean square error (RMSE) for all variant callers, concerning Ling’s results, it becomes apparent that *Musta* exhibits the lowest RMSE, indicating the highest precision (Figure 2b). Interestingly, *Musta*’s precision is enhanced when employing all variant callers compared to the combinations offered by the options: –strict (run only Mutect2, LoFreq, Strelka2), –soft (VarScan2, VarDict, MuSe), and –fast (LoFreq, VarScan2, Strelka2). After this initial quantitative analysis, we turn to a qualitative analysis to check for agreement on the mutations detected. The Venn diagram in Figure 2c highlights the concordance between *Musta* and Ling analysis to be nearly 90%. From the same diagram analysis, it emerges that *Musta* identified 137 variants unseen by Ling analysis, while there are only 30 mutations identified in the original study not captured by *Musta*. The robustness and reliability of *Musta* are further underscored by Figure 2d, where it is evident that nearly 99% of the somatic mutations were identified by at least four out of six Variant Callers, indicating not only the comprehensive coverage provided by *Musta*, but also the high quality of its results. This highlights the solidity of its ensemble approach in capturing a wide range of somatic mutations, thus confirming its reliability in the analysis of cancer samples.

#### 2.1.2. Classification

After identifying somatic mutations in tumor samples, the subsequent classification of these mutations is vital for understanding their biological significance and clinical implications, particularly in identifying driver mutations pivotal for cancer progression. *Musta* employs two variant annotation options: the Ensemble Variant Effect Predictor (VEP) and GATK’s Funcotator. Their results are summarized in Figure 3 and Appendix A, which also include detailed metrics such as the amount of non-swapped physical and virtual memory used, the number of MB read and written, and CPU usage time, providing a comprehensive overview of the resource consumption.

One striking difference between the two variant annotation tools is their runtime efficiency (see Figure 3a,b). VEP completes annotation for a single sample in approximately 15 min, demonstrating rapid processing. In contrast, Funcotator requires significantly more time, with an average runtime of over seven hours per sample. This substantial difference may influence the choice of annotation tool based on specific analysis requirements and available computational resources. When comparing the quantitative classification outcomes (Figure 3c,d), Funcotator tends to identify slightly fewer genes compared to VEP. However, Funcotator provides a richer variety of classifications per gene, albeit with only a marginal difference. This suggests that while Funcotator may identify fewer genes overall, it offers more detailed classifications for each identified gene.

Furthermore, a qualitative analysis reveals that both tools classify the same set of genes as frequently mutated genes (FLAG), as shown in Figure 4a,b, indicating a high level of agreement in identifying genes relevant to hepatocellular carcinoma (HCC). These findings are consistent with previous studies in the literature, which have already identified these top genes as strongly correlated with hepatocellular carcinoma [10,11,12,13,14,15,16,17,18]. This convergence reinforces the reliability of both tools for cancer analysis. Given the high level of agreement and robustness observed, the choice between VEP and Funcotator ultimately depends on user preferences. Factors such as runtime requirements, computational resources, familiarity with each tool’s functionalities, and ease of access to additional files may influence the decision-making process.

#### 2.1.3. Interpretation

In this section, the outcomes of the Interpretation module are examined (refer to Appendix A for details on non-swapped physical and virtual memory usage, MB read and written, and CPU usage time), focusing on the comparative analysis between the Hepatocellular Carcinoma (HCC) and the Liver Hepatocellular Carcinoma (LIHC) datasets [19] obtained from “The Cancer Genome Atlas” repository. The aim is to ascertain the concordance and consistency of results across these datasets, notwithstanding inherent disparities in sample size and diversity.

The comparison outcomes are illustrated in Figure 5, Figure 6 and Figure 7 However, before delving into the interpretation of the results, it is essential to acknowledge a significant inherent bias: the LIHC dataset comprises 365 samples from distinct patients, whereas the HCC dataset consists of only 23 samples from a single patient, representing a singular case of hepatocellular carcinoma. Consequently, it is expected that the LIHC results exhibit greater diversity and wider distribution compared to those of HCC.

An initial positive observation is evident from Figure 5a,b, where the gene *TTN* emerges as the most mutated gene in both datasets. This finding aligns with existing literature [10], which strongly associates *TTN* mutations with Hepatocellular Carcinoma. It is notable that in the HCC dataset, genes are mutated across all samples, whereas in the LIHC dataset, the number of samples with mutated genes varies, reflecting the diversity inherent in a cohort of multiple patients.

The decisive evidence comes from the subsequent plots, which compare the Mutational Signatures extracted from the two datasets, from the SBS databases. In both datasets, the first signature is SBS22, associated with exposure to aristolochic acid and significantly correlated with Hepatocellular Carcinoma (Figure 6a,b), As expected, three distinct signatures were extracted from the LIHC dataset, while only one was extracted from the HCC dataset.

This aligns with the observations in Figure 7a,b, where the LIHC dataset shows a more uniform distribution of Transition and Transversion ratios, whereas the HCC dataset exhibits a clear predominance of T>A mutations, overshadowing the others during signature extraction.

In conclusion, the comparative analysis reveals notable concordance between the HCC and LIHC datasets, despite differences in sample size and diversity. As we have often pointed out, and it is still worth reiterating, these findings enhance our understanding of the mutational landscape of Hepatocellular Carcinoma and underscore the importance of robust interpretation methods in genomic analysis.

### 2.2. Evaluating the Scalability and Portability of *Musta*

*Musta* leverages Docker, Snakemake, and Conda to ensure maximum scalability and portability. Tested on CentOS 7.9.2009, Ubuntu 20.04 and, later, Windows 11 Professional (with WSL 2 and Docker Desktop) and macOS, *Musta* demonstrates comprehensive cross-platform compatibility.

For scalability, *Musta* adopts a conservative approach to core management, always leaving one core free to prevent overloads. This requires a minimum dual-core system. Performance, memory usage, and storage utilization are influenced by the tools in the Detection and Classification modules and the number of samples. On an 4-core, 16 GB RAM system, the Detection module runs effectively with the ‘–fast’ option, and the Classification module with VEP annotations. Systems with 16–128 GB RAM and 4–56 cores can incorporate more variant callers and annotators. *Musta*’s flexibility allows users to add variant caller results without regenerating VCFs, enhancing resource management and user convenience. However, intermediate files can occupy up to 50% of the total size of input BAM files, so adequate disk space is necessary.

Finally, an internet connection is essential for downloading packages, updates, and dependencies, ensuring smooth installation and operation of *Musta*.

## 3. Discussion

*Musta*, both in tests and routine analysis, has proven to be a robust and flexible pipeline for accurate detection and comprehensive analysis of somatic variants in cancer. Its ease of installation and setup enables users, even without specific skills in computational programming, to process cancer data using a single command line. Furthermore, *Musta* encourages and simplifies the definition of protocols, thus adhering to an analysis style that adopts best data processing practices [8], and thereby enabling the transfer and reproducibility of analysis and results.

Currently, *Musta* is being utilized for cancer sample data analysis at the CRS4-NGS Core. Its reliability has been extensively tested on the Liver Hepatocellular Carcinoma (LIHC) dataset [19] obtained from TCGA Research Network (https://www.cancer.gov/tcga, accessed on 21 July 2024). Each cohort comprises hundreds of samples from distinct patients. Additionally, *Musta* has been tested on published data from the genome archive of the Beijing Institute of Genomics, including 23 liver cancer samples from the same patient. All software  tools incorporated into *Musta* have been carefully chosen after an in-depth literature review and practical testing, with the selection guided by the need to meet reliability, reproducibility, and confidence in results, while also ensuring ease of use. The Dockerized version of *Musta* ensures consistent and reproducible results across multiple platforms, making it a valuable tool for project collaborations. Moreover, *Musta* promotes the good practice of standardization, precise protocol definition, and seamless transferability to other users, ensuring consistency and reliability across analyses.

In summary, *Musta* represents a significant advancement in the field of somatic variant analysis in cancer, offering a reliable, flexible, and easily reproducible approach for researchers engaged in precision oncology research.

## 4. Materials and Methods

*Musta* is a comprehensive end-to-end pipeline developed to streamline the detection, classification, and interpretation of somatic mutations in cancer samples. *Musta*’s foundation is a user-friendly Python-based command-line tool that easily handles matched tumor-normal samples and that simplifies the analysis process for researchers, regardless of their programming expertise. Researchers can initiate the entire analysis by executing a single command line, making *Musta* accessible and efficient even for those without specific programming skills.

### 4.1. Overview

The *Musta* framework efficiently organizes cancer sample processing tools into three distinct analysis modules: detection, classification, and interpretation (Figure 8). Researchers have the flexibility to run each module independently, or combine all three modules for a more comprehensive analysis workflow. The *Musta* framework is designed with a layered architecture (Figure 9): the core is a Snakemake-based workflow [20], encapsulated in a Python framework and running in a Docker container [21]. A user-friendly Command-Line Interface (CLI) enables users to issue commands and provide input data.

The source code and the latest version of *Musta* framework is freely available at https://github.com/next-crs4/musta (accessed on 21 July 2024). A simple Makefile bootstraps *Musta*, taking care of the installation, configuration and running modules and allowing the execution of the entire pipeline or any individual module depending on the starting data. The detection module accepts BAM files as input, the classification module works with VCF files and the interpretation module accepts MAF files.

By running make bootstrap, the *Musta* Docker image is built and the executable file musta is conveniently linked in the user’s bin path. With this setup, you can easily access *Musta* commands using the musta executable. The command musta --help will show a brief usage help.

The basic structure of the *Musta* command is as follows:


musta COMMAND --workdir WORKING-DIR --samples-file SAMPLES-FILE [options]


Here is an overview of these components:COMMAND: This can be one of the following: detect, classify, or interpret. It selects the specific module the user wants to run. The command musta COMMAND --help will show a brief usage help for each command.--workdir WORKING-DIR: This parameter designates the working directory, which is the destination folder for analysis. This is the folder where (i) the Snakemake pipeline will be deployed, (ii) analysis logs will be stored, and (iii) all analysis outputs will be generated.--samples-file SAMPLES-FILE: this parameter points to a YAML file that lists the datasets the user wants to analyze.

For detailed usage instructions and user options, including the complete directory structure of the destination folder and the required YAML structure for the datasets file, please refer to the *Musta* documentation available at https://next-crs4.github.io/musta (accessed on 21 July 2024).

#### 4.1.1. Snakemake Core

At the heart of *Musta*’s architecture resides a Snakemake-based workflow. Snakemake, a powerful and versatile workflow content manager, offers modularity, scalability, and reproducibility. Within the Snakemake environment, the workflow is orchestrated through a set of rules that efficiently connect input datasets to their corresponding outputs. Snakemake smartly schedules rules, ensuring they run only when their necessary input files are available. It can also handle multiple rules simultaneously, making it scalable on different systems, allowing users to efficiently run the workflow on local machines, including high-performance computing clusters and cloud-based platforms. Furthermore, Snakemake exhibits resilience by offering error recovery capabilities. In the event of an interruption during pipeline execution due to an error, Snakemake identifies missing output files, empowering users to resume the execution of failed jobs from the last available correct results, once the issues are resolved. Another notable advantage of Snakemake is its ability to manage software dependencies using Conda. This implies that Snakemake can create dedicated independent virtual environments in which to execute analysis pipelines, ensuring that all required software dependencies are present and in the correct versions. In the context of Snakemake, the *Musta* workflow consists of two core files:Snakefile: This file represents the workflow’s core. It contains all the rules and commands that define the sequence of tasks and dependencies within the pipeline. Each rule specifies how to create output files from input files or other rules, and may invoke commands, scripts, or generate output directly. The Snakefile essentially orchestrates the entire analysis process.config.yaml: The config.yaml file complements the Snakefile by providing essential input files and parameters for the workflow. It specifies the data sources, settings, and configurations needed for *Musta* to execute the analysis correctly. This file helps to configure the workflow according to the user’s specific requirements and data. Furthermore, through boolean flags in the config.yaml, allow users to specify which module to execute and which tools to enable.

The *Musta* Snakemake workflow is freely available at https://github.com/solida-core/musta (accessed on 21 July 2024). Users who are experienced with Snakemake can run the workflow independently, outside of the *Musta* framework, by editing the Snakefile and the config.yaml and running Snakemake commands.

#### 4.1.2. Python framework

Through the integration of the Snakemake API library [22], the Python tool serves as the conductor of the Snakemake pipeline. It effectively manages configuration, orchestrates execution, and interprets user commands received through the CLI. In the following is reported an overview of the Python tool’s functionality:Pipeline download. Initially, it downloads the pipeline into the working directory (WORKING-DIR).Environment configuration. The tool sets up internal paths within the Docker container environment.Edit samples.yaml. It edits the samples.yaml file, replacing user-local paths with Docker volume paths.Edit config.yaml. It manipulates the config.yaml file to control execution by toggling module and tool flags and constructing the workflow of rules.Workflow execution. Finally, this tool initiates the Snakemake workflow, ensuring the execution of the defined tasks.

In summary, the Python tool acts as the orchestrator of the Snakemake pipeline, configuring the environment, handling file paths, and executing the workflow based on user-defined parameters and commands received through the CLI.

#### 4.1.3. Docker Container

*Musta* is conceived for an easy installation on any computational platform and any operating system through the Docker platform. Within the *Musta* architecture, the Docker container acts as a boundary layer, separating the outer CLI layer from the inner Python framework. This segregation allows a clear distinction between user interaction and core functionality. Key to this setup is the use of Docker volumes. Both the input data and the working directory are mounted as Docker volumes, providing access to the Python framework inside the container. Importantly, this arrangement is entirely transparent to the user, who will continue to work with familiar local file paths. This approach simplifies data management and ensures that results are readily accessible for further analysis. As previously mentioned, the make bootstrap command facilitates the creation of the *Musta* Docker image. Users can confirm the existence of this image by executing the docker image command. For users familiar with Docker, direct access to the *Musta* Python tool is possible using a command structured as follows:


docker run [docker volumes options] musta:Dockerfile musta [musta option]


However, managing Docker volume mounts manually can be cumbersome and time-consuming. To streamline this operation, we have developed a user-friendly CLI that resides at a higher layer in the *Musta* framework.

#### 4.1.4. Command Line Interface

The Command Line Interface (CLI) is based on a Bash script and aims to provide a seamless user experience by abstracting the presence of the Docker container, allowing users to interface directly with the Python tool. Here is a breakdown of the CLI’s functionality:Input file and dataset check. The CLI performs an initial check on input files and datasets. It ensures that they exist, are in the correct format, and that all required accessory files are present. For example, it verifies that BAM files are indexed (with corresponding BAI files for each BAM), checks that VCF files are in compressed (GZ) format and indexed, and confirms the completeness of reference files.Constructing the Docker command. After verifying input files, the CLI constructs the Docker command. For each input file, volumes are mounted in Docker, and the list of arguments for the *Musta* Python tool is assembled. Finally, the Docker RUN command is prepared.

The CLI streamlines the process by handling these tasks automatically, making it easier for users to work with *Musta* and Docker without the need for detailed Docker knowledge. It simplifies the interaction with the *Musta* Python tool, allowing users to focus on their analysis rather than Docker container management.

### 4.2. Workflow Modules

The workflow for detection and analysis of somatic mutations in cancer usually begins with the preprocessing of raw sequencing data and culminates in the identification of somatic mutations. The goal is to generate a comprehensive and annotated list of mutations that facilitates informed clinical decision-making [8]. Although there is currently no universally accepted bioinformatics pipeline or set of tools for cancer analysis, several common elaboration steps are typically required and widely accepted. Downstream analysis includes, among others, driver gene identification, pathway analysis, deconvolution of mutational signature analysis, and estimation of tumor heterogeneity [5,23].

The preprocessing step aims to remove low-quality reads, adapter sequences, and other artifacts from the raw sequencing data, because overall they could affect downstream analyses. This step involves quality control, trimming, filtering, and alignment to a reference genome [24,25,26,27]. To simplify *Musta*, we have omitted the preprocessing step from the pipeline. Users are encouraged to generate their preprocessed BAMs, following the GATK Best Practices [27]. For user convenience, we recommend our dedicated Snakemake pipeline available at https://github.com/solida-core/dima (accessed on 21 July 2024). This resource will guide users through the necessary steps for preprocessing their data effectively.

#### 4.2.1. Detection

The process of identifying somatic mutations in cancer begins with the comparison of aligned tumor sequencing data to a corresponding normal or matched reference dataset. This comparison makes it possible to distinguish between mutations that are tumor-specific (somatic mutations) and those present in the patient’s germline (germline mutations). Variants are subsequently filtered according to several criteria such as read depth, quality score, allele frequency and functional annotation, to remove false positives and retain true positives while minimizing false negatives [28].

The landscape of available somatic variant detection tools is very broad, including both commercial and open-source solutions [29,30,31,32,33]. However, the concordance among these tools about variant callers is often low [34].Dramatically, different somatic variant callers may yield divergent results when applied to the same dataset. This is because each caller has its own strengths and limitations [29,30,31,32,33], thus resulting in such a low concordance among them. Consequently, identifying a single best variant caller that can consistently outperform others on different datasets is impractical [31,32]. To address this issue, ensemble approaches have been introduced to harmonize the results generated by multiple somatic variant callers [35,36,37,38]. To create an effective ensemble approach for somatic mutation detection [39], two key issues need to be addressed:1.Selecting diverse and accurate component callers. Choosing an optimal number of component callers while ensuring diversity is crucial. Base learners must balance high accuracy with diversity to build a robust ensemble [40,41]. Diversity is essential because the benefit of an ensemble diminishes if all callers perform similarly. Conversely, too much diversity can lead to contradictory results, so a balanced selection of diverse component callers is essential.2.Combining individual caller results:(a)Simple approaches like majority voting [42] and consensus approaches [35,43] rely on majority decisions or consensus among individual caller results. However, these do not take into account the quality of each caller’s results, although weighted approaches are possible.(b)Complex machine learning-based methods such as stacking, Bayesian approaches, decision trees, and deep learning [44,45,46,47,48,49] leverage prediction results or metrics from individual callers as input features. Machine learning algorithms are used to combine these features, offering increased robustness against noise and errors. However, these methods often demand more computational resources and may be less interpretable.

In summary, the selection of diverse and accurate component callers, as well as the choice between simple and complex ensemble methods, plays a key role in improving somatic mutation detection accuracy while taking into account computational complexity and interpretability. After an initial in-depth literature review, practical testing, and screening, we selected six commonly used somatic variant callers: MuTect2 (v4.3.0) [50], VarScan2 (v2.4.4) [51], VarDict (v2019.06.04) [52], Strelka2 (v2.9.10) [53], LoFreq (v2.1.5) [54], and MuSE (v1.0) [55]. To improve the sensitivity and specificity of somatic mutation detection by leveraging the strengths of each caller, we selected SomaticSeq (v3.8.0) [47] as the ensemble approach for the *Musta* framework.

SomaticSeq integrates the VCF outputs from these six variant callers and processes them to produce a single consensus VCF, thus providing a more accurate and reliable set of somatic mutations. Compared to other ensemble and consensus tools—such as SomaticCombiner [56], NeoMutate [46], BAYSIC [45] and Moss [57]—SomaticSeq stands out for its consistent updates and maintenance, as well as its unique ability to accept input not only from all six variant callers selected for the *Musta* framework, but also from JointSNVMix [58], SomaticSniper [59], and Scalpel [60]. Additionally, it offers the flexibility to input any arbitrary VCF file(s) from caller(s) that we did not explicitly incorporate, paving the way for future improvements to *Musta*. These characteristics contribute to a richer, more flexible, and user-adaptive pipeline.

While technically feasible to develop a pipeline based on the other aforementioned ensemble and consensus tools, such an approach presents challenges that could compromise the universality and accessibility we aimed to achieve with *Musta*, without offering significant advantages. Each of these other tools has strengths and relevance to specific aspects of variant calling. For example, SomaticCombiner integrates a new variant allelic frequency (VAF) adaptive majority voting approach, which maintains sensitive detection for variants with low VAFs. NeoMutate incorporates seven supervised machine learning (ML) algorithms to exploit the strengths of multiple variant callers using a non-redundant set of biological and sequence features. BAYSIC performs an unsupervised, fully Bayesian latent class analysis to estimate false positive and false negative error rates for each input method. Moss, in addition to VCF files, also takes the original BAM files of the tumor and normal samples as input. However, the downstream analysis flexibility achieved with SomaticSeq is far more effective for a general approach to the problem. Overall, SomaticSeq’s adaptability and comprehensive input acceptance make it the superior choice for developing a robust and versatile variant calling pipeline, enhancing both the utility and ease of use for researchers.

SomaticSeq uses an ensemble strategy based on machine learning to integrate the individual caller results. It employs a random forest classifier that takes into account various features extracted from each variant caller’s output, such as allelic depths, base quality scores, and mapping qualities. These features are used to train the classifier to distinguish true somatic mutations from false positives. The trained classifier then combines the predictions of the component callers and generates a final set of high-confidence somatic mutations. SomaticSeq into our analysis pipeline, will improve the reliability and completeness of somatic mutation analysis in cancer genomes.

In the context of Snakemake workflow, the Detection module involves rules that, starting from BAM files, automatically performs variant calling in tumor-normal matched mode, depending on provided input files for each sample. The rules are responsible for efficiently preparing the input files for each variant caller selected, executing the variant calling and the ensembles processes and retrieving the resulting VCF files. This includes handling tasks such as format conversions and file compression and decompression, as needed, and the indexing of the resulting VCF files for further downstream analysis. The basic structure of the *Musta* command to invoke Detection module is:


musta detect --workdir WORKING-DIR --samples-file SAMPLES-FILE [options]


By default, the detection module within *Musta* is configured to perform the variant calling using all six of the mentioned variant callers. However, *Musta* provides users with the flexibility to customize their analysis according to their specific needs in the following ways:Excluding specific variant callers. If users wish, *Musta* allows specific variant callers to be excluded from the analysis. This customization ensures that only the desired variant callers are effectively applied to the data.Selecting preset combinations. Alternatively, users can choose from preset combinations of variant callers. *Musta* offers predefined sets of variant callers that have been optimized for specific analysis scenarios:--strict run only restrictive variant callers: Mutect2, LoFreq, Strelka2.--soft run only permissive variant callers: VarScan2, VarDict, MuSe.--fast run only fast variant callers: LoFreq, VarScan2, Strelka2, MuSe.

#### 4.2.2. Classification

Following the variant calling process, where somatic mutations are identified, it is imperative to annotate these variants with functional information to gain insights into their potential impact on the genome. At a fundamental level, gene annotation is performed to determine whether the variant affects the protein-coding sequence of a gene. It specifically identifies whether the variant is synonymous or non-synonymous, and assesses its impact on splice sites and other specific genomic regions. This initial gene-level annotation is crucial for understanding the potential functional consequences of the variant itself. To enhance the annotation process, the identified variants are compared and annotated using established and widely used databases, such as dbSNP [61], gnomAD [62], ClinVar [63], and COSMIC [64]. These databases house a vast repository of curated genomic information, enabling researchers to determine whether a particular variant has been previously reported in the germline or has been associated with cancer-related genomic changes. This comparison is critical in assessing the significance of the variant and its potential relevance to the specific tumor or tumor-associated disease under study.

To streamline the annotation process, researchers have several widely used and freely available annotation tools at their disposal. These tools, such as the Ensemble Variant Effect Predictor (VEP) [65], ANNOVAR [66], SnpEff [67], and Funcotator from GATK, play a key role in annotating variants. They use diverse annotation algorithms and leverage comprehensive genomic resources to provide detailed functional annotations for the identified variants. With the help of these tools, researchers can efficiently annotate large-scale datasets, taking into account variant type, location, and potential functional impact. In a comparative study [68], these variant annotation tools were evaluated using a ground-truth set of 298 variants from a medical exome database. Notably, VEP exhibited the highest accuracy, annotating 297 variants with HGVS nomenclature. In contrast, ANNOVAR showed the highest discrepancy, with 20 variants showing inconsistencies. Another study [69] reported a robust 92.6% concordance (100/108 variants) between SnpEff and VEP.

Based on the findings of these comparative studies, we have selected VEP (v106) as the primary variant annotator for the *Musta* pipeline. However, to enhance the flexibility and robustness of our pipeline, we recognize the unique capabilities and advantages of other annotation tools. Therefore, we have integrated Funcotator (v4.3.0) as an option within the pipeline, allowing users to choose and compare annotations generated by both VEP and Funcotator. This approach empowers users to tailor their annotation strategy to their specific research needs and preferences, ensuring that *Musta* can handle a variety of datasets with different characteristics. Additionally, we are committed to further expanding the annotation capabilities of *Musta*. The roadmap for a future release includes the integration of SnpEff, providing users with even more flexibility and options in their analyses. By incorporating multiple annotation tools, *Musta* aims to offer a comprehensive, adaptable, and user-friendly solution for somatic variant annotation, capable of accommodating diverse datasets and evolving research requirements.

By default, the classification module in *Musta* is configured to perform variant annotation using VEP, but users have the flexibility to opt for Funcotator alone or in combination with VEP. The fundamental structure of the *Musta* command for invoking the Classification module is:


musta classify --workdir WORKING-DIR --samples-file SAMPLES-FILE [options]


After annotating somatic mutations using both VEP and Funcotator, the resulting annotated Variant Call Format (VCF) files are further transformed into the Mutation Annotation Format (MAF). In fact, the MAF format is accepted as the standard one for storing both somatic and germline variants stemming from extensive cancer sequencing studies [70]. Once the variants are in the MAF format, the pipeline moves to the final stage of interpreting these variants through downstream analyses. These analyses are vital for unraveling the functional significance and potential clinical implications of the annotated somatic mutations.

#### 4.2.3. Interpretation

The downstream analysis of somatic mutation data commonly involves the utilization of multiple independent software tools, employing various computational and statistical approaches. In the *Musta* pipeline, this analysis is significantly facilitated by incorporating the Maftools R package (v2.10) [71]. Maftools is a specialized bioinformatics tool designed for the comprehensive analysis and interpretation of somatic variants stored in the MAF format. It offers a wide range of downstream analysis and visualization modules that are extensively used in cancer genomic studies, enabling researchers to perform driver gene identification, pathway analysis, mutational signature analysis, enrichment analysis, and association analysis, among others [5,23]. Moreover, Maftools addresses the challenge of visualizing complex and heterogeneous data, providing researchers with an array of visualization functions to generate publication-quality images such as oncoplots, lollipop plots, and oncoprints. By incorporating Maftools into the *Musta* pipeline, we offer to researchers the possibility to effectively interpret somatic variants, derive meaningful insights, conduct in-depth analyses, and communicate their findings more efficiently, ultimately enhancing our understanding of the genomic landscape in cancer research.

1.Variant visualization. Within the *Musta* pipeline framework, Maftools presents a range of graphical representations that assist in detecting mutation patterns and recurring characteristics within the dataset.Summary plots provide an at-a-glance summary by showcasing variant counts per sample and their distribution based on classification. They offer a high-level overview of the mutation landscape within the dataset.Onco plots illustrate mutations across samples, revealing distribution patterns. They are definitely valuable in cancer patient studies, providing a comprehensive mutational view.Lollipop plots take the interpretation a step further by depicting mutations on protein structures. This visual representation helps researchers to gain a better understanding of the precise locations of mutations and their potential impact on protein structure and thus function.Transitions and transversions plots categorize mutations into transitions and transversions, providing insights into the mutational spectrum within the dataset. Understanding these mutation types is crucial for unraveling mutational patterns.Rainfall plots are a powerful tool for visualizing mutation-rich areas within cancer genomes. They are especially useful for identifying hypermutated regions, a phenomenon known as kataegis [23,72]. These plots help pinpoint areas of intense genomic alteration.Oncostrip allows researchers to zoom in on specific genes, simplifying the exploration of features like mutual exclusivity. This focused view aids in uncovering relationships and interactions among genes of interest.2.Somatic interactions. Recent advances in cancer genomics research have shown that many disease-causing genes in cancer are often mutated in a mutually exclusive manner [73,74]. Identification of such gene sets can reveal de novo pathways and underlying mechanisms of tumorigenesis. For this, *Musta* performs a Fisher’s exact test on all combinations of genes, to detect such mutually exclusive or co-occurring sets of genes.3.Detecting cancer driver genes. Cancer driver genes provide selective growth advantage to cancer cells when mutated [3]. Several mathematical approaches have been developed to identify such driver genes [5,75,76,77]. In the context of the *Musta* framework, detection of such associated genes is based on the oncodriveCLUST algorithm [78]: the concept is that a majority of the mutations in oncogenes are clustered around mutational hotspots, whereas mutations on passenger genes are randomly distributed.4.Pfam domains. In each type of cancer, specific protein domains are notably enriched with mutations [79,80]. The process of identifying and categorizing protein domains based on their mutation frequency serves the dual purpose of discerning the predominant domain affected within a particular cancer cohort. This approach further aids in pinpointing highly disrupted pathways and protein families that share similar functions, offering insights into the intricacies of deregulated mechanisms.5.Tumor heterogeneity. Tumors are generally heterogeneous, composed of multiple clones and undergoing continuous evolution [81]. Heterogeneity can be inferred by clustering and classifying variants into sub clones, according to their allele frequencies [82,83]. Although clustering of variant allele frequencies gives us a fair idea on heterogeneity, it is also possible to measure the extent of heterogeneity in terms of a numerical value. MATH score is a simple quantitative measure of intra-tumor heterogeneity, which expresses the width of the variant allele frequency (VAF) distribution [84]. High MATH scores are found to be associated with poor prognosis and survival [85].6.Mutational signature analysis. As cancer progresses, it develops a characteristic mutational pattern that unveils the underlying mutagenic processes at play. This revealing pattern can be deciphered through dimensional reduction techniques like non-negative matrix factorization (NMF) [23]. By decomposing a matrix containing nucleotide substitutions categorized into 96 substitution classes, based on their surrounding bases, specific mutational signatures, unique to each cancer type, emerge. These mutational signatures offer a blurred picture of the intricate mutational landscape of cancers. However, by cross-referencing them with validated signatures, we can achieve a clearer and more focused picture. This further enriches our understanding of the distinct processes that drive cancer progression.

## Figures and Tables

**Figure 1 ijms-25-08044-f001:**
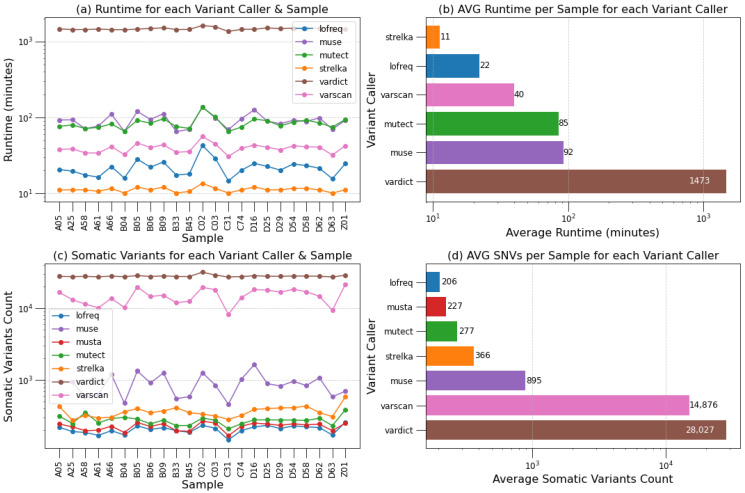
Performance evaluation of *Musta*’s detection module on HCC datasets. This figure provides a comprehensive comparison of the variant calling results from the *Musta* Detection module and the original study by [9]. (**a**,**b**) display the execution times for each variant caller across different samples: (**a**) highlighs the runtime of each variant caller for each sample while (**b**) shows the average runtime for a single sample. (**c**,**d**) illustrate the total counts of *pass* variants reported by each variant caller: (**c**) exhibits the number of variants called by each variant caller for each sample while (**d**) shows the average number of called variants in a single sample per variant caller.

**Figure 2 ijms-25-08044-f002:**
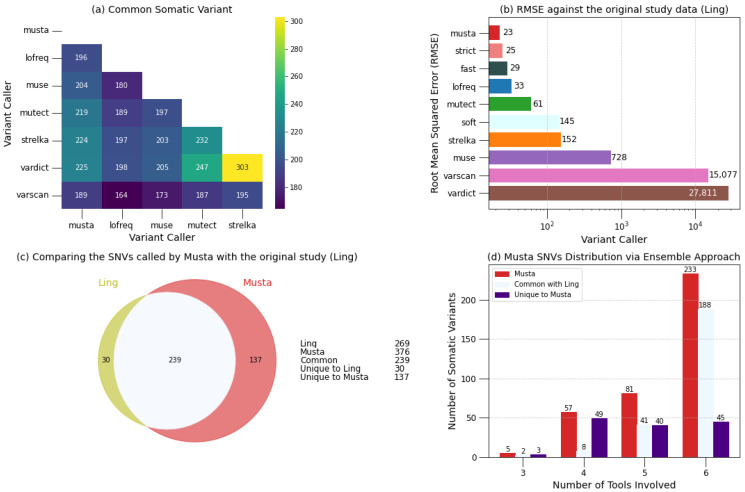
*Musta* against original study data (Ling [9]). This figure shows a comparison of the number of somatic variants validated by Ling and the results from *Musta*. (**a**) heatmap highlights common somatic variants between variant callers and their contribution in *Musta* results. (**b**) root mean square error (RMSE) for all variant callers against Ling results, demonstrating *Musta*’s precision. (**c**) Venn diagram highlighting the concordance between *Musta* and Ling analysis, showing nearly 90% overlap, with *Musta* identifying 137 unique variants and Ling 30. (**d**) underscores *Musta*’s robustness, showing that nearly 99% of somatic mutations were identified by at least four out of six variant callers, indicating the high quality and comprehensive coverage of *Musta*’s ensemble approach.

**Figure 3 ijms-25-08044-f003:**
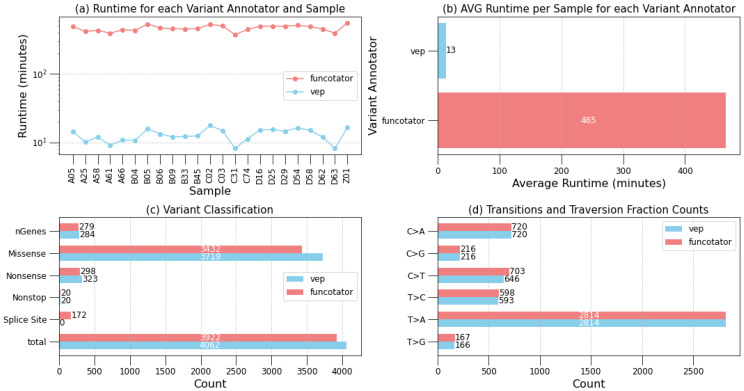
Performance evaluation of *Musta*’s classification module on HCC datasets. This figure summarizes the results and efficiency of the Ensemble Variant Effect Predictor (VEP) and GATK’s Funcotator used in the *Musta* framework. (**a**,**b**): Runtime efficiency comparison. VEP annotates a sample in 15 min, while Funcotator takes over 7 h. (**c**,**d**): Quantitative outcomes. Funcotator identifies slightly fewer genes than VEP, but offers more detailed classifications per gene.

**Figure 4 ijms-25-08044-f004:**
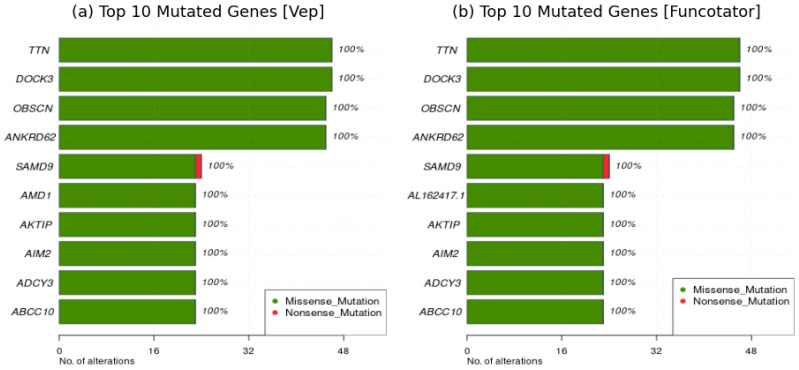
Comparison of frequently mutated genes (FLAG) classified by VEP and Funcotator. Both tools classify the same set of genes as frequently mutated genes (FLAG), indicating high agreement in identifying genes relevant to hepatocellular carcinoma (HCC).

**Figure 5 ijms-25-08044-f005:**
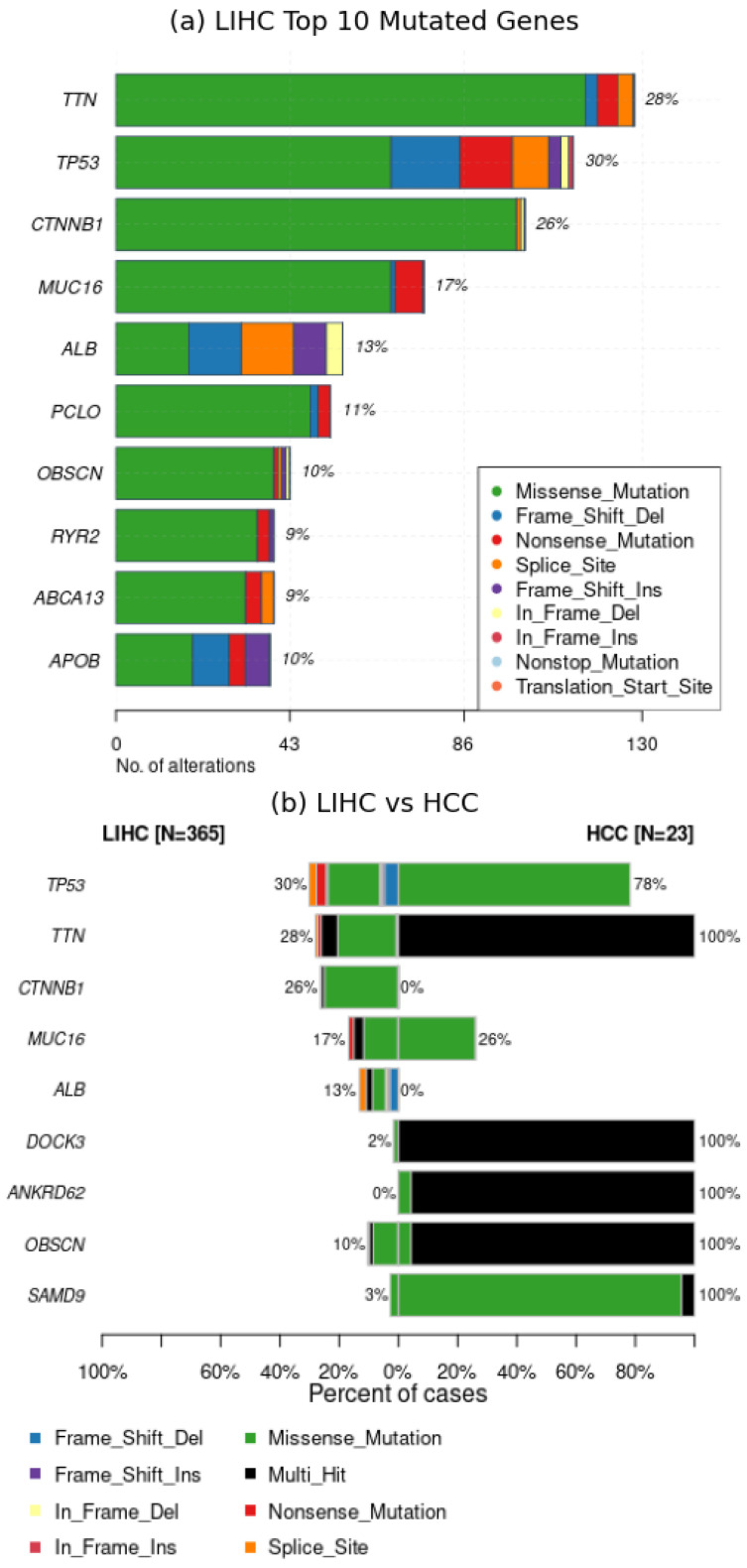
Performance evaluation of *Musta*’s interpretation module on HCC and LIHC datasets: frequently mutated genes. The (**a**) shows the Top 10 mutated genes in TCGA-LIHC dataset, while (**b**) illustrates a comparison between Top 10 mutated genes in TCGA-LIHC and HCC (Ling) datasets. *TTN* gene is identified as the most mutated gene in both datasets, aligning with literature on Hepatocellular Carcinoma. The HCC dataset shows mutations in all samples, while the LIHC dataset exhibits variable numbers of samples with mutations, reflecting greater diversity.

**Figure 6 ijms-25-08044-f006:**
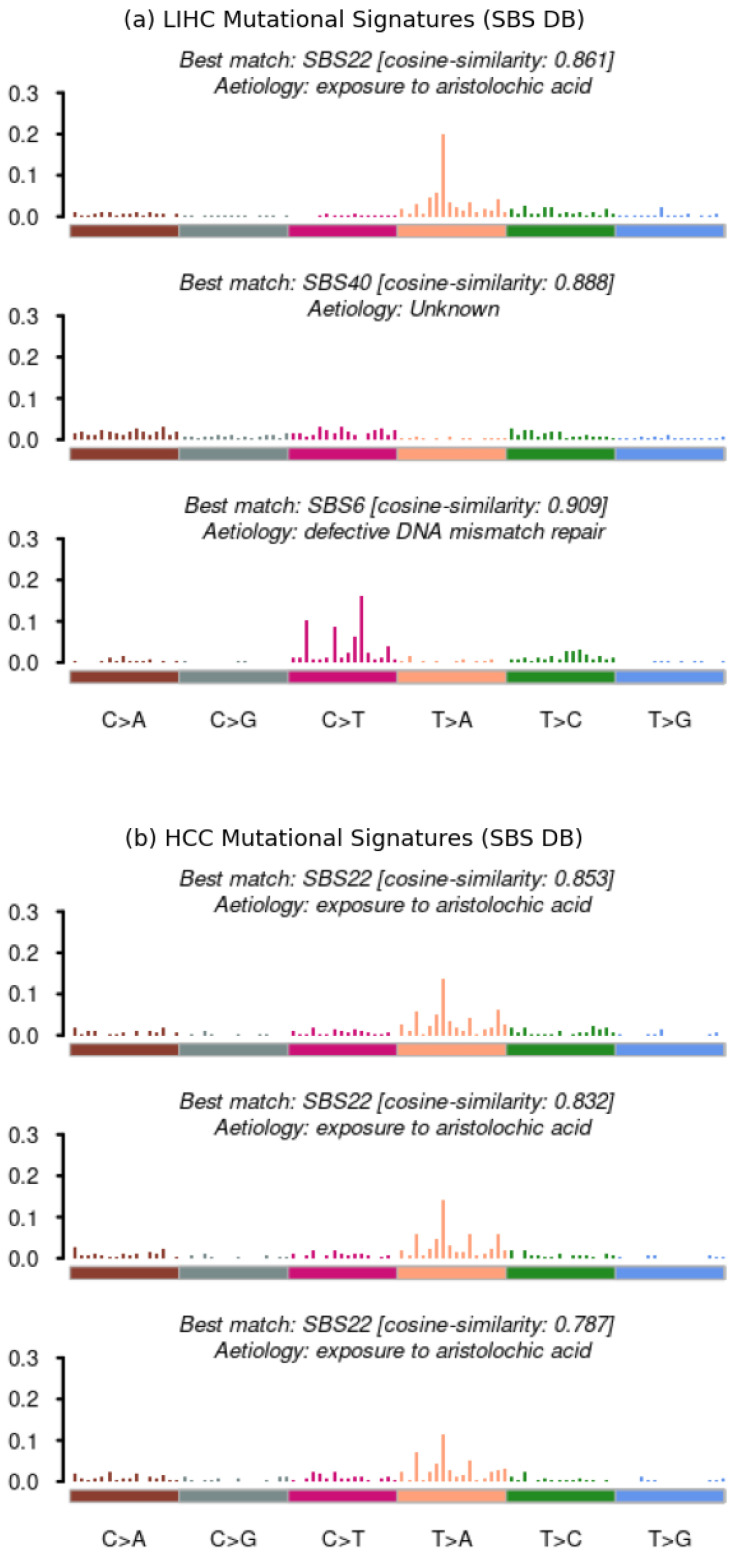
Performance evaluation of *Musta*’s interpretation module on HCC and LIHC datasets: mutational signatures from the SBS databases. Both datasets feature SBS22, linked to Hepatocellular Carcinoma, as the first signature.

**Figure 7 ijms-25-08044-f007:**
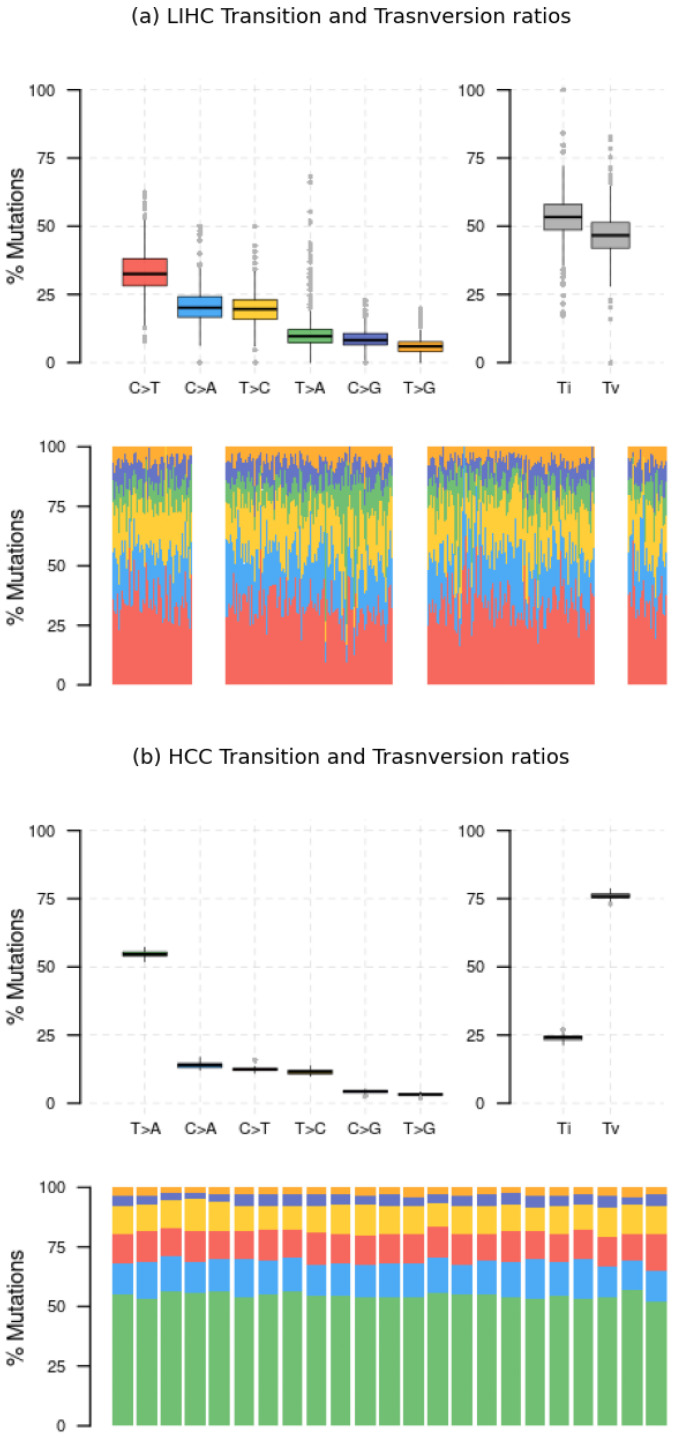
Performance evaluation of *Musta*’s interpretation module on HCC and LIHC datasets: transition and trasnversion ratios. Transition and transversion ratios are more uniformly distributed in the LIHC dataset. In contrast, the HCC dataset shows a predominance of T>A mutations.

**Figure 8 ijms-25-08044-f008:**
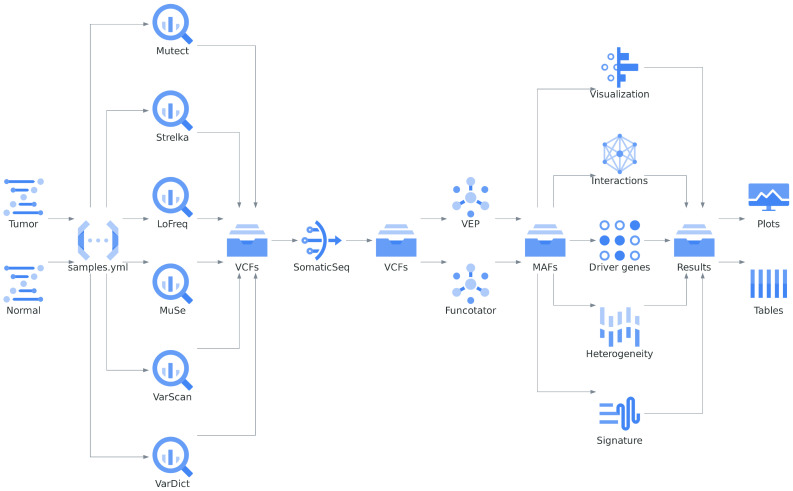
Overview of the *Musta* framework for cancer sample analysis. This figure illustrates the workflow of the *Musta* framework, which efficiently organizes cancer sample processing tools into three distinct analysis modules: detection, classification, and interpretation. The process begins with input BAM files, which are detailed in the samples.yaml file. Each paired normal and tumor BAM file is sent to one of the six variant callers in the detection module. The VCF files generated by each variant caller are then processed by the Ensemble step (SomaticSeq), which produces a consensus VCF. This consensus VCF is subsequently passed to the Classification module, where two Variant Annotators (VEP and Funcotator) generate an annotated MAF file. This annotated MAF file serves as the input for the final step, the Interpretation module, which generates plots and tables to facilitate data analysis and visualization.

**Figure 9 ijms-25-08044-f009:**
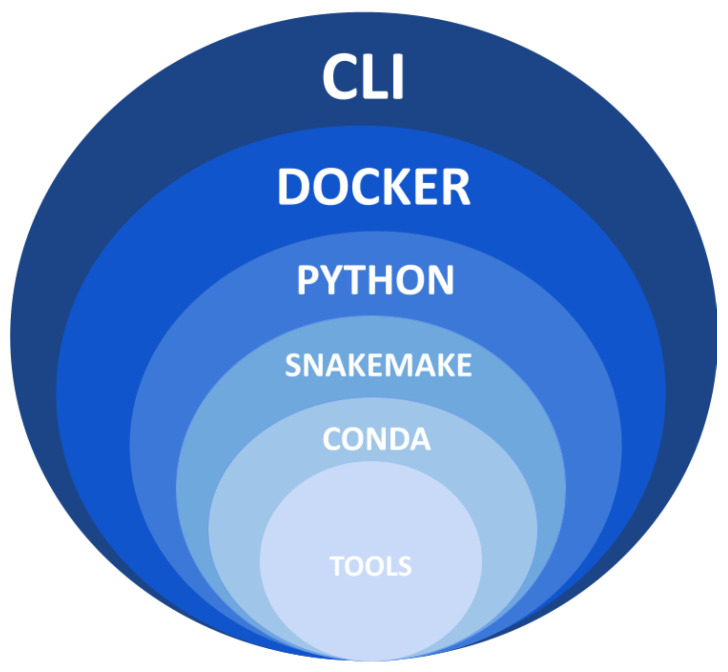
Layered architecture of the *Musta* framework. The *Musta* framework is built with a layered architecture, ensuring efficient and organized processing of cancer samples. At its core is a Snakemake-based workflow (version 7.15), encapsulated within a Python framework (version 3.8) and executed in a Docker container (version 20 and later). The Snakemake rules instantiate the necessary Conda (version 4.12) environments to run the individual tools, ensuring compatibility and reproducibility. Users interact with the system through a user-friendly Command-Line Interface (CLI), enabling easy command execution and data input.

## Data Availability

The HCC datasets analyzed for this study can be found at the genome sequence archive of Beijing Institute of Genomics under accession id PRJCA000091: https://bigd.big.ac.cn/bioproject/browse/PRJCA000091 (accessed on 21 July 2024). *Musta* is publicly released at the following GitHub repository: https://github.com/next-crs4/mustapy.

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
