# Peer review of "A Novel Affordable and Reliable Framework for Accurate Detection and Comprehensive Analysis of Somatic Mutations in Cancer"

_ijms, 2024, doi:10.3390/ijms25158044_

Round 1

Reviewer 1 Report

Comments and Suggestions for Authors

In the manuscript, the authors present a comprehensive and user-friendly pipeline for detecting, classifying, and interpreting somatic mutations in cancer. The workflow integrates multiple third-party tools and leverages Snakemake and Docker for management and installation. The authors applied large cancer datasets to demonstrate the effectiveness of the workflow. However, there are a few areas that can be improved before  publication:

1.      System Requirements: As a workflow, the authors should describe the operating system and computational configuration requirements. For example, the minimum number of CPUs and memory size required for optimal performance.

2.      Resource Usage: It would be beneficial to add a resource usage report based on test data, including storage, memory, CPU usage, and running times for each module. This information would help users estimate the resources needed for their analyses.

3.      Figure Formatting: Please ensure that the font formats are consistent across all figures. Some figures appear shrunken, such as Figures 4d and 4e. Additionally, the font size is too small in Figures 5c and 5d, making them difficult to read.

4.      Figure Clarity: In Figure 1, a black line should be removed at the bottom. Additionally, please make Figure 1 clearer to improve its readability and presentation quality.

Comments on the Quality of English Language

The quality of English looks good to me.

Author Response

  1. System Requirements: As a workflow, the authors should describe the operating system and computational configuration requirements. For example, the minimum number of CPUs and memory size required for optimal performance.

Authors’ response: We thank the reviewer for the suggestion. We have implemented this in the revised version of the manuscript, adding the new subsection 3.2 "Evaluating the scalability and portability of Musta." We have also, in the Results section, added this sentence, "The server runs CentOS Linux release 7.9.2009, with Docker version 20.10.11 and GNU Make version 4.3."

  1. Resource Usage: It would be beneficial to add a resource usage report based on test data, including storage, memory, CPU usage, and running times for each module. This information would help users estimate the resources needed for their analyses.

Authors’ response: Following the reviewer's suggestion, we have modified the manuscript as follows:

- We have updated supplementary tables S1 and S4 and added a new table S5, in which we report the amount of non-exchanged physical and virtual memory used, the number of MB read and write, and the CPU utilization time.

- In subsection 3.1.1 (Detection) we added:

"In addition, supplementary tables S1 include data on the amount of non-exchanged physical and virtual memory used, the number of MB read and written, and the CPU utilization time"

"It is important to note the variation in execution time:" Strelka, Lofreq and Varscan complete their analysis in a few tens of minutes, Mutect and Muse take about an hour and a half, and Vardict takes over 24 hours. Excluding Vardict, the Detection module typically takes about 4 hours to analyze a single sample."

- In subsection 3.1.2 (Classification) we added:

"which also include detailed metrics such as the amount of non-exchanged physical and virtual memory used, the number of MB read and write, and CPU usage time, providing a complete overview of resource consumption."

- In subsection 3.1.3 (Interpretation) we added:

"(refer to Supplementary Table S5 for details on non-exchanged physical and virtual memory utilization, MB read and write, and CPU utilization time)."

  1. Figure Formatting: Please ensure that the font formats are consistent across all figures. Some figures appear shrunken, such as Figures 4d and 4e. Additionally, the font size is too small in Figures 5c and 5d, making them difficult to read.

Authors’ response: We redrew all figures using the same fonts and modified Figures 3, 4 and 5. To improve readability and resolution, we divided the original Figure 3 into two new figures (Figures 3 and 4), the original Figure 4 into two figures (Figures 5 and 6), and the original Figure 5 into three new figures (Figures 7, 8, and 9). Figure captions have been updated.

  1. Figure Clarity: In Figure 1, a black line should be removed at the bottom. Additionally, please make Figure 1 clearer to improve its readability and presentation quality.

Authors’ response: We have removed the black line in Figure 1 and improved the figure in terms of readability and clarity as requested.

Reviewer 2 Report

Comments and Suggestions for Authors

In this paper, the authors developed a Snakemake-based pipeline for detecting somatic mutations in cancer data by incorporating previously published methods. The manuscript includes a detailed description of the command line, which is very helpful for readers who do not have much programming experience. Although the manuscript is well-written and includes essential details, there are some missing parts in the methodology. I suggest that the authors revise the manuscript with considerable effort to make the paper more robust.

Major concerns:

1.      From line 213 to 215. The author said that different methods have different strengths and limitations, this is true since different types of cancer have very different characteristics. To establish a comprehensive method is very important, but how to keep the specific strengths into the method is a challenge. As described in the manuscript, SomaticSeq was used for somatic mutation detection. However, why this method was chosen are not well described. I would like to suggest incorporating other detection method as options in the pipeline to make this tool more robust to handle different scenarios.   

2.      Similar situation in 2.2.2 classification. Since the tool is a pipeline that can incorporating other method, only use one method probably not good enough to handle different datasets.

Minor concerns:

1.      Figure 4 and 5 are not readable.

Author Response

Major concerns:

  1. From line 213 to 215. The author said that different methods have different strengths and limitations, this is true since different types of cancer have very different characteristics. To establish a comprehensive method is very important, but how to keep the specific strengths into the method is a challenge. As described in the manuscript, SomaticSeq was used for somatic mutation detection. However, why this method was chosen are not well described. I would like to suggest incorporating other detection method as options in the pipeline to make this tool more robust to handle different scenarios.

Authors’ response: We thank the reviewer for giving us the opportunity to better clarify this aspect.

Our selection of SomaticSeq stems from its unique combination of useful features, which collectively make it the optimal choice for achieving the goals of our research. Despite some significant technical challenges, we will plan to integrate options for different mutation detectors into the pipeline. We will incorporate tools that have proven effective in research and are well-maintained, enhancing the robustness and versatility of our approach. We aim to ensure that our pipeline remains at the cutting edge of variant detection technology, providing researchers with the best possible tools to address their specific needs. Integrating additional mutation detectors will also enable us to leverage the strengths of various tools for specific aspects of variant calling, each one excelling in certain scenarios.

In particular, we added the following text in subsection 2.2.1 (Detection) of the revised version of the manuscript “SomaticSeq integrates the VCF outputs from these six variant callers and processes them to produce a single consensus VCF, thus providing a more accurate and reliable set of somatic mutations. Compared to other ensemble and consensus tools - such as SomaticCombiner [37], NeoMutate [35], BAYSIC [34] and Moss [37] - SomaticSeq stands out for its consistent updates and maintenance, as well as its unique ability to accept input not only from all six variant callers selected for the Musta framework but also from JointSNVMix [48], SomaticSniper [49] and Scalpel [50]. Additionally, it offers the flexibility to input any arbitrary VCF file(s) from caller(s) that we did not explicitly incorporate, paving the way for future improvements to Musta.These characteristics contribute to a richer, more flexible, and user-adaptive pipeline. While technically feasible to develop a pipeline based on the other aforementioned ensemble and consensus tools, such an approach presents challenges that could compromise the universality and accessibility we aimed to achieve with Musta, without offering significant advantages. Each of these other tools has strengths and relevance to specific aspects of variant calling. For example, SomaticCombiner integrates a new variant allelic frequency (VAF) adaptive majority voting approach, which maintains sensitive detection for variants with low VAFs. NeoMutate incorporates seven supervised machine learning (ML) algorithms to exploit the strengths of multiple variant callers using a non-redundant set of biological and sequence features. BAYSIC performs an unsupervised, fully Bayesian latent class analysis to estimate false positive and false negative error rates for each input method. Moss, in addition to VCF files, also takes the original BAM files of the tumor and normal samples as input. However, the downstream analysis flexibility achieved with SomaticSeq is far more effective for a general approach to the problem. Overall, SomaticSeq’s adaptability and comprehensive input acceptance make it the superior choice for developing a robust and versatile variant calling pipeline, enhancing both the utility and ease of use for researchers.”

  1. Similar situation in 2.2.2 classification. Since the tool is a pipeline that can incorporating other method, only use one method probably not good enough to handle different datasets.

Authors’ response: We thank the reviewer once again for giving us the opportunity to better explain the procedures we adopted. In the revised version of the manuscript, we have added the following text to subsection 2.1.2 (Classification): “However, to enhance the flexibility and robustness of our pipeline, we recognize the unique capabilities and advantages of other annotation tools. Therefore, we have integrated Funcotator as an option within the pipeline, allowing users to choose and compare annotations generated by both VEP and Funcotator. This approach empowers users to tailor their annotation strategy to their specific research needs and preferences, ensuring that Musta can handle a variety of datasets with different characteristics. Additionally, we are committed to further expanding the annotation capabilities of Musta. The roadmap for a future release includes the integration of SnpEff, providing users with even more flexibility and options in their analyses. By incorporating multiple annotation tools, Musta aims to offer a comprehensive, adaptable, and user-friendly solution for somatic variant annotation, capable of accommodating diverse datasets and evolving research requirements.”.

 Minor concerns:

  1. Figure 4 and 5 are not readable.

Authors’ response: We redrew all figures using the same fonts and modified figures 3, 4, and 5 to improve their readability. We split original Figure 3 into two new figures (Figures 3 and 4), the original Figure 4 into two figures (Figures 5 and 6), and the original Figure 5 into three new figures (Figures 7, 8, and 9). Figure captions have been updated.

Round 2

Reviewer 2 Report

Comments and Suggestions for Authors

The authors answered my question, and I endorsed it for publication.